# Nonperturbative effects in triple-differential dijet and Z+jet production at the LHC

Stefan Gieseke ®[3], Maximilian Horzela ®[1], Manjit Kaur ®[4], Dari Leonardi[3], Klaus Rabbertz ®[2*], Aayushi Singla ®[4], Cedric Verstege ®[2]

[1]II. Institute of Physics, Georg-August Universität Göttingen, , Göttingen, 37077 Germany.
[2]Institute of Experimental Particle Physics, Karlsruhe Institute of Technology, Wolfgang-Gaede-Str. 1, Karlsruhe, 76131, Germany.
[3]Institute of Theoretical Physics, Karlsruhe Institute of Technology, Wolfgang-Gaede-Str. 1, Karlsruhe, 76131, Germany.
[4]Department of Physics, Panjab University, Sector-14, Chandigarh, 160014, India.

*Corresponding author(s). E-mail(s): klaus.rabbertz@kit.edu;
Contributing authors: stefan.gieseke@kit.edu; maximilian.horzela@uni-goettingen.de; manjit@pu.ac.in; aayushi.singla@cern.ch; cedric.verstege@kit.edu;

**Abstract**

In comparisons of precision collider data to the most accurate highest-order calculations in perturbative quantum chromodynamics (QCD), it is required to correct for nonperturbative effects. Such effects are typically studied using Monte Carlo event generators that complement fixed-order predictions with perturbative parton showers and models for the nonperturbative effects of the Underlying Event and hadronisation. Thereby, the final state of collision events can be predicted at the level of stable particles, which serve as input for full detector simulations.

This article investigates the impact of nonperturbative effects on two processes that may be used for precision determinations of the strong coupling constant and the proton structure: the triple-differential dijet and Z+jet production. While nonperturbative effects impact both processes, significant differences among them are observed and further investigated. Indications are found that the Underlying Event and hadronisation cannot fully explain these differences and the perturbative modelling may play a significant role as well.

**Keywords:** QCD, nonperturbative, dijet, Z+jet

## 1 Introduction

To match the unprecedented precision of measurements at the LHC, most accurate predictions of at least next-to-next-to-leading order (NNLO) in perturbative QCD (pQCD) are required. An indispensable ingredient for such calculations are parton distribution functions (PDFs) that describe the manifestly nonperturbative (NP) internal structure of the colliding hadrons and in particular of the proton. Despite significant progress in lattice gauge theory it is not yet possible to derive the proton structure from first principles such that the proton PDFs must be determined in dedicated fits of PDF-dependent observables to precise measurements. Moreover, further NP effects from

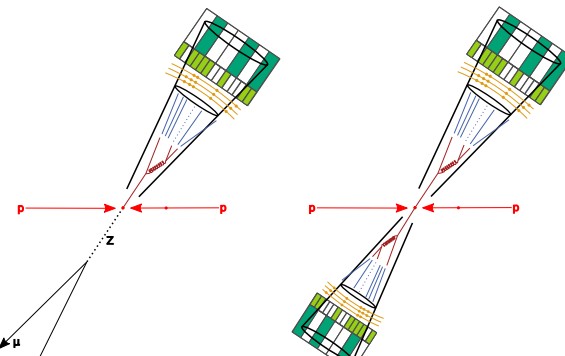

**Fig. 1**: Illustrations of Z+jet (left) [11] and dijet production (right). In the Z+jet case, an oppositely-charged muon pair and at least one jet are required. For the dijet case, a second jet replaces the muon pair.

hadronisation and the Underlying Event (UE) must be estimated by means of Monte Carlo (MC) event generators, since typically the highest-order predictions in pQCD only provide events at parton level.

In this article, NP effects for two processes in proton–proton collisions at the LHC are studied: the inclusive production of dijet and Z+jet events. For both processes, illustrated in Fig. 1, predictions are available at NNLO accuracy in pQCD [1–5]. While baseline fits of proton PDFs rely on deep-inelastic scattering (DIS) [6], which to first degree depends on the valence quarks and only indirectly on the strong coupling $\alpha_s$, dijet and Z+jet production both are directly sensitive to $\alpha_s$ and the gluon content of the proton [7–10]. Measurements of these processes at the LHC therefore hold significant potential towards improving $\alpha_s$ and the gluon PDF by resolving ambiguities in their determination arising from correlations.

Following the publications by the CMS Collaboration on dijet production at the LHC [12, 13] we choose a setup where the cross section is investigated triple-differentially as a function of the two angular variables $y_\mathrm{b}$ and $y^*$ plus one quantity with dimension of energy (or momentum) that correlates with the energy scale of the collision. The two angular variables are defined as

$$y_\mathrm{b} = \frac{1}{2}\,|y_1 + y_2| \qquad (1)$$

and

$$y^* = \frac{1}{2}\,|y_1 - y_2| \qquad (2)$$

where $y_1$ and $y_2$ are the rapidities of the two leading-$p_\mathrm{T}$ objects emerging from a collision, i.e. either the leading two jets in dijet production or the reconstructed Z boson and the leading jet in Z+jet events. The quantity $y_\mathrm{b}$ corresponds to the longitudinal boost of the centre-of-mass with respect to the laboratory system of a $2 \to 2$ scatter, while $y^*$ is the rapidity of the two back-to-back final-state objects in the centre-of-mass frame. This choice is particularly useful, because the variables $y^*$ and $y_\mathrm{b}$ to first order separate the dependence on the partonic scattering angle in the centre-of-mass system from the dependence on the imbalance in the initial-state parton momenta, and hence the PDFs. Moreover, one can derive the following relations to the proton momentum fractions $x_{1|2}$ carried by the incoming partons of a $2 \to 2$ scatter:

$$\begin{aligned} x_{1|2} &= \frac{2\langle p_\mathrm{T}\rangle_{1,2}\,\cosh(y^*)}{\sqrt{s}}\,\exp(\pm y_\mathrm{b}) \\ &= \frac{m_{1,2}}{\sqrt{s}}\,\exp(\pm y_\mathrm{b})\,. \end{aligned} \qquad (3)$$

Here, $\langle p_\mathrm{T}\rangle_{1,2}$ and $m_{1,2}$ are the average transverse momentum, $\langle p_\mathrm{T}\rangle_{1,2} = (p_{\mathrm{T,jet1}}+p_{\mathrm{T,jet2}})/2$, and the invariant mass, $m_{1,2} = \sqrt{(\mathbf{p}_1 + \mathbf{p}_2)^2}$, of the leading two jets, where $p_{\mathrm{T,jet1}}$, $p_{\mathrm{T,jet2}}$ are their transverse momenta and $\mathbf{p}_1$, $\mathbf{p}_2$ their four-momenta, respectively. Since at leading order (LO) for dijet events $p_{\mathrm{T,jet1}} = p_{\mathrm{T,jet2}} = \langle p_\mathrm{T}\rangle_{1,2} = p_{\mathrm{T,jet}}$ and for Z+jet events $p_{\mathrm{T,Z}} = p_{\mathrm{T,jet}}$ we focus in the following on $\langle p_\mathrm{T}\rangle_{1,2}$ and $p_{\mathrm{T,Z}}$ as third variable, which we generically label as $X$. Results with respect to the dijet mass can be found in the Appendix A.

In the following, jets are always clustered with the collinear- and infrared-safe anti-$k_\mathrm{T}$ jet algorithm [14] with size parameters of $R = 0.4$ and $R = 0.8$. In the main part we focus on jets with $R = 0.4$. Results for $R = 0.8$ are presented as additional material in the Appendix A.

The triple-differential cross section in these three observables can be written in terms of weighted MC events as

$$\frac{d^3\sigma}{dX\,dy_\mathrm{b}\,dy^*} = \frac{N_\mathrm{eff}\,(\Delta X, \Delta y_\mathrm{b}, \Delta y^*)}{N_\mathrm{eff,tot}}\,\sigma_\mathrm{incl}\,, \qquad (4)$$

where $N_{\text{eff}}(\Delta X, \Delta y_{\text{b}}, \Delta y^*)$ denotes the effective number of selected events within a given phase space in $X$, $y_{\text{b}}$, and $y^*$; $N_{\text{eff,tot}}$ is the effective number of generated events in the full phase space and $\sigma_{\text{incl}}$ is the inclusive cross section for the analysed process. The effective number of events can be expressed in terms of event weights $w_i$ as

$$N_{\text{eff},\mathcal{S}} = \sum_{i \in \mathcal{S}} w_i \qquad (5)$$

of the events $i$ within a phase space region $\mathcal{S}$.

This article is organised as follows: Sect. 2 defines the phase space for the analyses. Non-perturbative corrections for the two investigated processes are derived and discussed in Sect. 3. The separation of NP effects into hadronisation and the Underlying Event is addressed in Sect. 4. A dedicated section 5 examines the behaviour of typical observables for UE studies in the context of the previous section. A summary of the main findings is presented in the concluding Sect. 6, while additional material not discussed in the main text is provided in the Appendix A.

## 2 Analysis phase space

The derived variables $y_{\text{b}}$ and $y^*$ of Eqs. 1 and 2 depend on the rapidities $y_1$, $y_2$ of the leading objects measured in a collision event. The latter are constrained by acceptance limitations in the experiments, e.g. ATLAS or CMS [15, 16], which can be expressed in terms of the coverage in pseudorapidity $\eta$ of the relevant subdetectors. For an optimal reconstruction, jets should be measured within coverage of the inner tracking devices extending typically up to $|\eta| \approx 2.5$. Muons can be measured by the ATLAS (CMS) detectors up to $|\eta| = 2.7$ (2.4), leading to a similar constraint on the rapidity $y_Z$ of the Z boson that is reconstructed from an opposite-sign muon pair. The transition from coordinate axes $(y_1, y_2)$ to $(y_{\text{b}}, y^*)$ corresponds to a rotation by $\pi/4$ in rapidity phase space. Applying cuts on $(y_1, y_2)$ leads, after mapping to the first quadrant only, to a triangular-shaped phase space in $(y_{\text{b}}, y^*)$, as illustrated in Fig. 2 for the Z+jet process. Here, we choose a binning scheme for $y_{\text{b}}$ and $y^*$ with a bin width of 0.5 starting from zero up to 2.5. The maximum absolute rapidity allowed for either $y_1$ or $y_2$

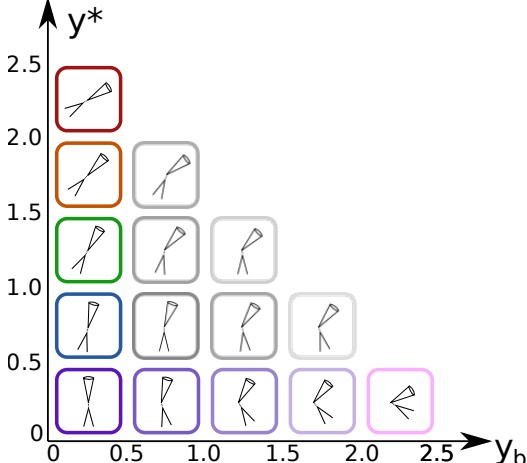

**Fig. 2**: Visualisation of the analysed Z+jet phase space in $y_{\text{b}}$ and $y^*$ as defined in Eqs. 1-2. The bins along the $y_{\text{b}}$ and $y^*$ axes discussed in the main part of this work are shown with coloured boxes.

is given by $y_{\text{b}} + y^* < |y|_{\text{max}} = 3.0$, which slightly larger than the tracker coverage.

This results in a total of 15 $(y_{\text{b}}, y^*)$ bins. In the region where $y_{\text{b}} < 0.5$, there are five bins ranging from $y^* < 0.5$ to $2.0 \leq y^* < 2.5$. Each following $y_{\text{b}}$ region has one $y^*$ bin less than the previous one, finishing with a single bin $y^* < 0.5$ for $2.0 \leq y_{\text{b}} < 2.5$. In the main part of this work, we focus on the nine bins corresponding either to a series of bins with increasing $y_{\text{b}}$ for $y^* < 0.5$, or with increasing $y^*$ for $y_{\text{b}} < 0.5$ to better showcase our observations. These bins are indicated by coloured boxes in Fig. 2, while results for the remaining bins, drawn with grey boxes, can be found in Figs. A1–A4 of the Appendix.

For the final selection criteria and binning of dijet events we follow the publication by CMS [13] and impose $|y|_{\text{max}} = 3.0$ for the two leading jets. In contrast to CMS, however, we reduce the $p_{\text{T,jet}}$ thresholds from $p_{\text{T,jet1}} > 100\,\text{GeV}$ and $p_{\text{T,jet2}} > 50\,\text{GeV}$ to $30\,\text{GeV}$ and $25\,\text{GeV}$ respectively. This allows for an increased overlap in phase space when comparing the dijet and Z+jet processes.

Since the production cross section drops drastically with increasing $\langle p_{\text{T}} \rangle_{1,2}$, the bin widths grow with $\langle p_{\text{T}} \rangle_{1,2}$ to accommodate for the lower expected number of events. Equally, towards outer rapidity the cross section becomes smaller. Therefore, bins in $\langle p_{\text{T}} \rangle_{1,2}$ are merged in nine edge

**Table 1**: The binning scheme in $\langle p_\mathrm{T}\rangle_{1,2}$ for the dijet analysis. Due to limited statistical precision, some bins are merged in the edge bins (E) as compared to the central bins (C) in the plane of $(y_\mathrm{b}, y^*)$.

| Binning scheme | binning labels in $(y_\mathrm{b}, y^*)$ plane |
|---|---|
| (cf. Fig. 2) | E<br>E  E<br>C  E  E<br>C  C  E  E<br>C  C  C  E  E |

| | bin edges in $\langle p_\mathrm{T}\rangle_{1,2}$ / GeV |
|---|---|
| Central (C) | 47, 57, 67, 77, 87, 97, 107, 127, 147, 175, 207, 243, 284, 329, 380, 437, 499, 569, 646, 732, 827, 931, 1046, 1171, 1307, 1458, 1621, 1806, 2003, 2217, 2453, 2702 |
| Edge (E) | 47, 57, 67, 77, 87, 97, 107, 127, 147, 207, 284, 380, 499, 646, 827, 1046, 1307 |

**Table 2**: The binning scheme in $p_\mathrm{T,Z}$ for the Z+Jet analysis. Due to limited statistical precision, some bins are merged in the edge bins (E) as compared to the central bins (C) in the plane of $(y_\mathrm{b}, y^*)$. An extra wide binning (X) is used for the highest $y^*$-bin.

| Binning scheme | binning labels in $(y_\mathrm{b}, y^*)$ plane |
|---|---|
| (cf. Fig. 2) | X<br>E  E<br>C  C  E<br>C  C  C  E<br>C  C  C  C  E |

| | bin edges in $p_\mathrm{T,Z}$ / GeV |
|---|---|
| Central (C) | 25, 30, 35, 40, 45, 50, 60, 70, 80, 90, 100, 110, 130, 150, 170, 190, 220, 250, 400, 1000 |
| Edge (E) | 25, 30, 35, 40, 45, 50, 60, 70, 80, 90, 100, 110, 130, 150, 170, 190, 250, 1000 |
| Extra (X) | 25, 30, 40, 50, 70, 90, 110, 150, 250 |

bins (E) at outer rapidities as compared to six central ones (C). The final binning scheme chosen for dijet events is presented in Table 1.

For the Z+jet process, the Z boson is reconstructed from a pair of oppositely charged muons, where the invariant mass of the dimuon system has to be within a range of $\pm 20\,\mathrm{GeV}$ around the Z-boson mass $m_\mathrm{Z} = 91.1876\,\mathrm{GeV}$ [17]. Muons are considered for an absolute pseudorapidity smaller than 2.4 and a transverse momentum of at least $25\,\mathrm{GeV}$. If multiple Z-boson candidates can be reconstructed, the one closest to $m_\mathrm{Z}$ is chosen. The transverse momentum of the reconstructed Z boson has to be at least $p_\mathrm{T,Z} > 25\,\mathrm{GeV}$. Jets are required to have a minimum $p_\mathrm{T}$ of $20\,\mathrm{GeV}$ and an absolute rapidity of less than 2.4. Any jet that overlaps with a prompt muon within $\Delta R(\mathrm{jet},\mu) < 0.3$ is omitted. The distance $\Delta R$ to the muon is defined as

$$\Delta R(\mu, \mathrm{jet}) = \sqrt{\left(\phi_\mathrm{jet} - \phi_\mu\right)^2 + \left(\eta_\mathrm{jet} - \eta_\mu\right)^2},$$

where $\phi$ and $\eta$ are the azimuthal angles and pseudorapidities of the corresponding jet and muon. This ensures that muons, which are also clustered to form jets, are removed from further consideration as "jet".

Similar to the dijet case, a binning scheme with bin widths increasing with $p_\mathrm{T,Z}$ and fewer bins in the outer rapidity regions is chosen. Moreover, the

suppression of the LO Z+jet process through PDF effects reduces the expected number of events at high $y^*$, where parton momenta of $x_1 \approx x_2$ are required. As a result an extra-wide binning (X) is employed for the $(y_\mathrm{b}, y^*)$ bin with ($y_\mathrm{b} < 0.5$, $2.0 \leq y^* < 2.5$). The final binning scheme chosen for Z+jet events is presented in Table 2.

# 3 Nonperturbative effects

We derive correction factors to fixed-order perturbative calculations for the presented dijet and Z+jet analyses to estimate the size and uncertainty of the effect introduced by the NP models in MC event generators. The full event generation until particle level usually passes through the following steps: the calculation of the perturbative scattering amplitudes and subsequent phase space integration (matrix element, ME), the collinear resummation of leading logarithms with parton showers (PS), the fragmentation of coloured final-state partons into hadrons and subsequent decays of unstable hadrons (Had), and the addition of UE activity via multiple-parton interactions (MPI). Here, the two steps labelled as Had and MPI are considered as NP effects. Hence, we define the NP correction to fixed-order predictions at parton level as the ratio between the nominal predictions with and without hadronisation effects and MPI.

The correction at a given perturbative order (PO) of the ME can then be written as:

$$C_{\mathrm{NP}}^{\mathrm{PO}} = \frac{\sigma_{\mathrm{ME+PS+Had+MPI}}^{\mathrm{PO}}}{\sigma_{\mathrm{ME+PS}}^{\mathrm{PO}}}, \qquad (6)$$

where the subscripts ME+PS+Had+MPI and ME+PS indicate, which steps of the MC event generation have been executed. In other words, the numerator represents the nominal cross section at the particle level given by any MC event generator, while the denominator is the cross section derived from the same MC generator with hadronisation effects and MPI turned off. Because the coloured partons after parton showers can interact with coloured partons stemming from the UE modelling via MPI (colour reconnection), the event generation process must be run twice, once for the numerator and a second time for the denominator leading to statistically independent event samples.

For an automated and scalable generation of the MC events and subsequent analysis, the MCRun software framework [18] based on the workflow management tool LAW [19] is used. The LAW package implements and calls the tasks related to the necessary code executions, automatically resolves their dependencies, and manages and distributes the computational payloads on available computing resources. The configurations used for the results in this work are made available within the MCRun software allowing for an efficient reproduction of the same workflow. A dedicated RIVET [20] routine for each process, respectively, analyses event records produced by the two MC event generators HERWIG7 version 7.2.3 [21, 22] and SHERPA version 2.2.15 [23]. Jets are clustered utilising the FASTJET [14, 24] library. Selection cuts based on reconstructed objects, as described in the previous Sect. 2, are applied, and fiducial cross sections for each differential bin are calculated. These cross sections are computed for event generations up to particle (full) and parton level only (partial generation). Finally, NP correction factors are calculated following Eq. 6 from the obtained cross sections for each bin.

The obtained correction factors $C_{\mathrm{NP}}$ are subject to statistical fluctuations due to the limited number of generated events in both the full and partial generation sequences. To smooth these fluctuations, the NP correction is parameterised using an interpolating function

$$f(x) = a \cdot \ln(x/\mathrm{GeV})^b + c, \qquad (7)$$

in each $(y_{\mathrm{b}}, y^*)$ bin.

This parameterisation is chosen, since the NP effects are expected to contribute in phase-space regions where the typical energy scale of the collision approaches $\Lambda_{\mathrm{QCD}}$ while fading out towards higher energy scales. Since the third observable is correlated with the energy scale of the collision process we expect to see larger NP effects towards lower values of $X$, while the NP correction factors should approach unity towards high $X$. Indeed, the fitted values for the parameter $c$ are compatible with unity and the obtained fits match the data well, supporting these expectations independent of the analysed topology, perturbative order, or $(y_{\mathrm{b}}, y^*)$ bin.

## 3.1 Dijet production

The triple-differential dijet cross section is measured as a function of the three observables $y_{\mathrm{b}}$ and $y^*$, as defined in Eqs. 1-2, and either the average transverse momentum $\langle p_{\mathrm{T}} \rangle_{1,2}$ or the mass $m_{1,2}$ of the dijet system. The resulting NP correction factors derived with HERWIG7 at LO and NLO [21, 22, 25, 26] accuracy in pQCD of the ME are shown in Fig. 3 for anti-$k_{\mathrm{T}}$ jets with $R = 0.4$ in the selected $(y_{\mathrm{b}}, y^*)$ bins.

Further bins can be found in the Appendix in Fig. A1 together with correction factors for dijet production vs. the dijet invariant mass $m_{1,2}$ of the leading and subleading jet. NP correction factors for anti-$k_{\mathrm{T}}$ jets with radius parameter $R = 0.8$ can be found as additional material in Fig. A2 of the Appendix.

The observed correction factors for dijet production are close to unity for large $\langle p_{\mathrm{T}} \rangle_{1,2}$ ($m_{1,2}$) and decrease towards small $\langle p_{\mathrm{T}} \rangle_{1,2}$ ($m_{1,2}$) as expected. They exhibit no significant dependence on the perturbative order or $(y_{\mathrm{b}}, y^*)$ bin.

## 3.2 Z+jet production

This process can be measured triple-differentially in a similar fashion as the dijet production. The observable that quantifies the scale of the hard interaction is chosen to be the transverse momentum of the muon pair, labelled as $p_{\mathrm{T,Z}}$ in the

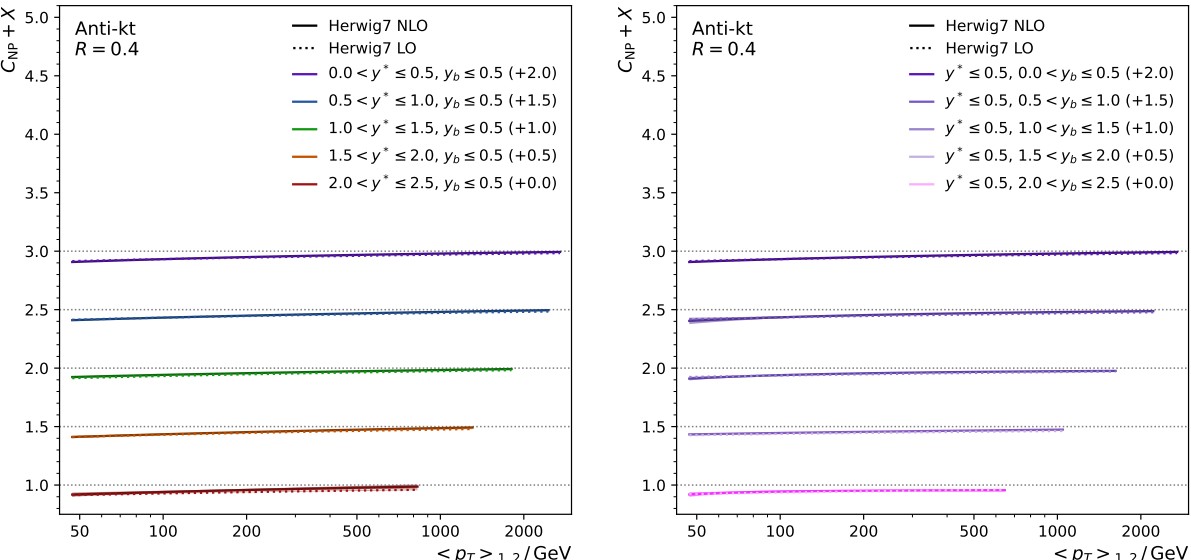

**Fig. 3**: $C_{\mathrm{NP}}$ vs. dijet $\langle p_{\mathrm{T}} \rangle_{1,2}$ using LO (dotted) and NLO MEs (solid lines) in HERWIG7 predictions for anti-$k_{\mathrm{T}}$ jets with $R = 0.4$. Bands indicate the statistical uncertainty. The fitted curves are shown for a series of phase space intervals increasing in $y^*$ for $y_{\mathrm{b}} < 0.5$ (left) and increasing in $y_{\mathrm{b}}$ for $y^* < 0.5$ (right). The factors are shifted with respect to unity by constant offsets.

following:

$$p_{\mathrm{T,Z}} = \left| \vec{p}_{\mathrm{T},\mu^-} + \vec{p}_{\mathrm{T},\mu^+} \right| , \qquad (8)$$

where $\vec{p}_{\mathrm{T},\mu^-}$ and $\vec{p}_{\mathrm{T},\mu^+}$ are the transverse momenta of the muon pair forming the $Z$-boson candidate. The rapidity observables $y_{\mathrm{b}}$ and $y^*$ are defined analogously to Eqs. 1-2 with $y_1$ being the rapidity of the Z boson, $y_{\mathrm{Z}}$, and $y_2$ corresponding to the rapidity of the leading jet in the event $y_{\mathrm{jet1}}$.

NP correction factors derived with HERWIG7 at LO and NLO accuracy in pQCD are presented in Fig. 4 for anti-$k_{\mathrm{T}}$ jets with $R = 0.4$ for the selected $(y_{\mathrm{b}}, y^*)$ bins.

Further $(y_{\mathrm{b}}, y^*)$ bins as well as all bins for anti-$k_{\mathrm{T}}$ jets with $R = 0.8$ can be found as additional material in Figs. A3–A4 of the Appendix.

In contrast to the behaviour observed for dijet production, we find a trend towards larger correction factors with rising $y^*$ but not with rising $y_{\mathrm{b}}$. This trend in $y^*$ seems to be somewhat reduced at NLO, where all NP correction factors exhibit an overall decrease in magnitude as compared to LO.

As a cross-check, the NP correction factors derived using HERWIG7 are compared to SHERPA [23, 27] at LO and NLO accuracy in Fig. 5.

The NP corrections derived with SHERPA are overall larger than the ones predicted by HERWIG7. The general features, however, of smaller corrections at NLO than at LO and the $y^*$ dependence are reproduced by SHERPA.

To verify the trend observed with increasing parton multiplicity at ME level, NP correction factors with multi-leg merged Z+jet with one, two, and three jets at NLO ME accuracy are generated using SHERPA. The corresponding NP correction factors are shown in Fig. 6.

Again a similar dependence on $y^*$ is visible in the NP correction factors derived from the multi-leg merged Z+jet event sample. Also, the behaviour of overall smaller NP corrections with higher multiplicity in the ME for the event generation is confirmed. Consistently, no dependence on $y_{\mathrm{b}}$ is observed.

This dependence of the NP correction factors on $y^*$ and on the perturbative order of the MEs in generating the hard amplitudes indicates that part of the observed NP correction factors might, in fact, originate in the perturbative modelling. The labelling as "nonperturbative" might therefore not be completely appropriate. Furthermore, the fact that this is only observed for the Z+jet but not

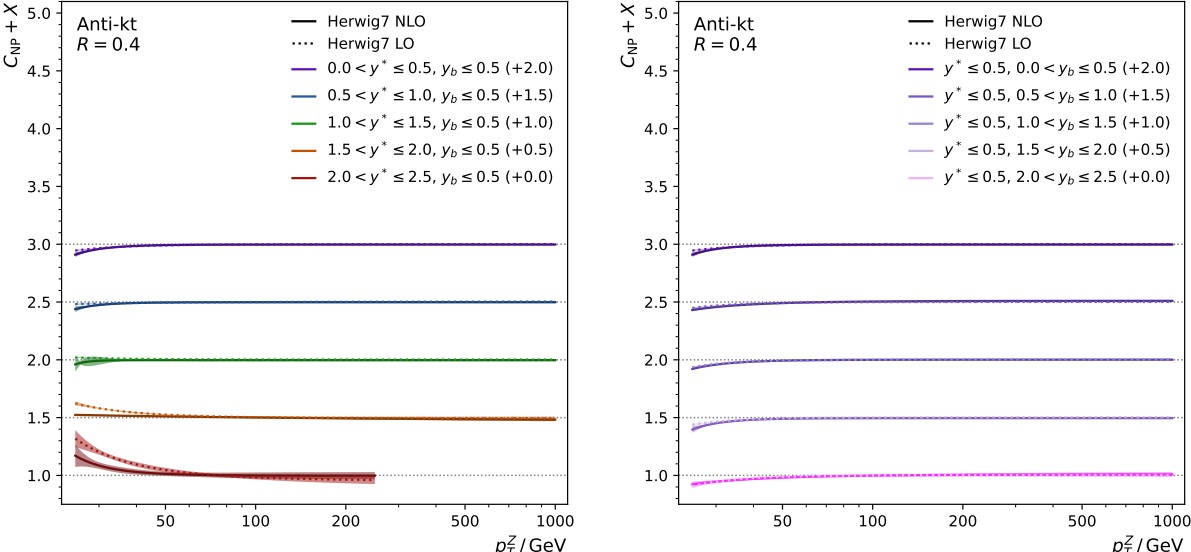

**Fig. 4**: Same as Fig. 3 but vs. $p_{\mathrm{T,Z}}$ in Z+jet production.

for the dijet process motivates a more thorough investigation.

# 4 Hadronisation & MPI Effects

According to our definition the NP correction factors account for two effects, hadronisation and the UE, which we attempt to study individually in the following.

The parton final state is determined by the parton showers that produce additional, visible activity in all hard events. This also holds for the additional events from MPI that are superimposed with the primary hard events. After the parton showers, all partons in the final state are assigned a colour line, which in turn assigns a colour partner to each parton. This would lead to uncorrelated events from MPI and to a strong growth of charged particle multiplicity with the number of additional hard events. However, the observation of increasing average transverse momentum in higher multiplicity events necessitates colour reconnection (CR) models that lead to interference between the coloured final states of the primary hard events and the MPI events.

The modelling of colour reconnection is based on the idea of colour pre-confinement, i.e. the observation that partons that are close to each other in phase space (usually momentum space) tend to exchange soft, NP gluons in a way that their colour lines are reconnected to become partners with close neighbours. While there are ideas to put this on a firm theoretical basis of a colour evolution model [28–30], most modern event generators still model this effect by minimising a "colour distance" of neighbouring partons [31–33]. For our consideration, which is purely based on observations of the final state, we should simply keep in mind that colour reconnection also links the MPI with the hadronisation models that we took as signposts for the NP modelling.

In order to nonetheless get an approximate, individual handle on the hadronisation and MPI effects, respectively, we compute hadronisation and MPI correction factors similar to Eq. 6 as

$$C_{\mathrm{Had}}^{\mathrm{PO}} = \frac{\sigma_{\mathrm{ME+PS+Had}}^{\mathrm{PO}}}{\sigma_{\mathrm{ME+PS}}^{\mathrm{PO}}} \qquad (9)$$

and

$$C_{\mathrm{MPI}}^{\mathrm{PO}} = \frac{\sigma_{\mathrm{ME+PS+MPI}}^{\mathrm{PO}}}{\sigma_{\mathrm{ME+PS}}^{\mathrm{PO}}} \quad . \qquad (10)$$

The hadronisation correction factors $C_{\mathrm{Had}}$ for Z+jet production with anti-$k_{\mathrm{T}}$ jets with radius parameter $R = 0.4$ computed with Herwig7 at LO and NLO accuracy in pQCD of the ME are shown in Fig. 7. They are consistently approaching

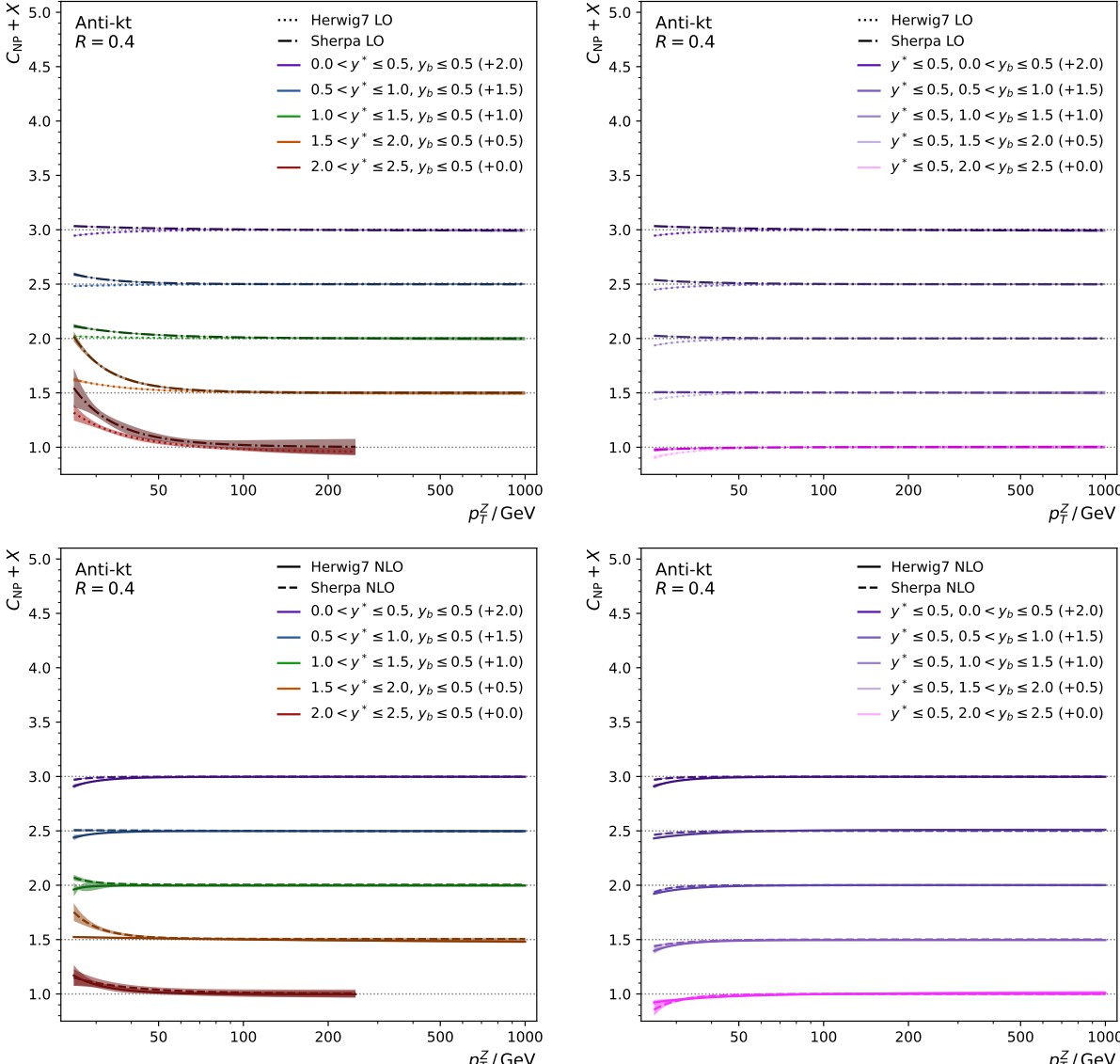

**Fig. 5**: Comparison of $C_{NP}$ vs. $p_{T,Z}$ between HERWIG7 (dotted & solid lines) and SHERPA predictions (dash-dotted & dashed lines) using LO (top row) and NLO MEs (bottom row) for anti-$k_T$ jets with $R = 0.4$. Bands indicate the statistical uncertainty. The fitted curves are shown for a series of phase space intervals increasing in $y^*$ for $y_b < 0.5$ (left) and increasing in $y_b$ for $y^* < 0.5$ (right). The factors are shifted with respect to unity by constant offsets.

a value of unity for high $p_{T,Z}$. Towards small $p_{T,Z}$ the downwards deviation from unity increases as expected for out-of-cone effects through hadronisation. However, they show no significant dependence on $y_b$, $y^*$ or the perturbative order of the generated samples.

On the contrary, the corresponding MPI correction factors $C_{MPI}$ for Z+jet production computed with HERWIG7, shown in Fig. 8, reveal trends versus $y^*$ and the perturbative order resembling the ones observed for the NP correction factors in Sect. 3. They also approach unity towards high $p_{T,Z}$, while for small $p_{T,Z}$ this time

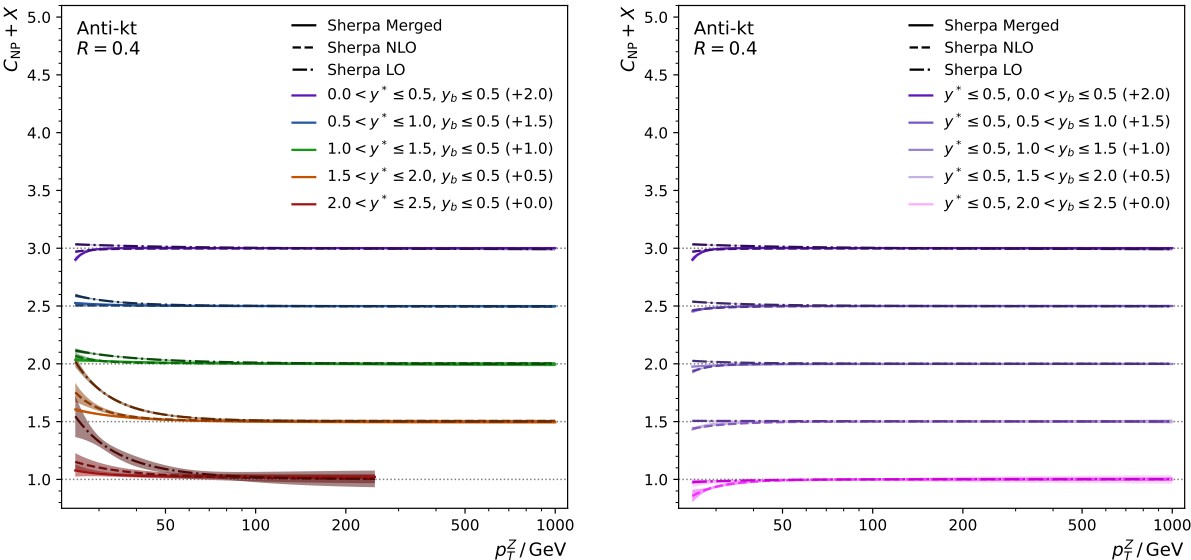

**Fig. 6**: Comparison of $C_{\mathrm{NP}}$ vs. $p_{\mathrm{T,Z}}$ among SHERPA predictions using LO (dash-dotted), NLO (dashed), and multi-leg merged MEs (solid lines) for anti-$k_{\mathrm{T}}$ jets with $R = 0.4$. Bands indicate the statistical uncertainty. The fitted curves are shown for a series of phase space intervals increasing in $y^*$ for $y_{\mathrm{b}} < 0.5$ (left) and increasing in $y_{\mathrm{b}}$ for $y^* < 0.5$ (right). The factors are shifted with respect to unity by constant offsets.

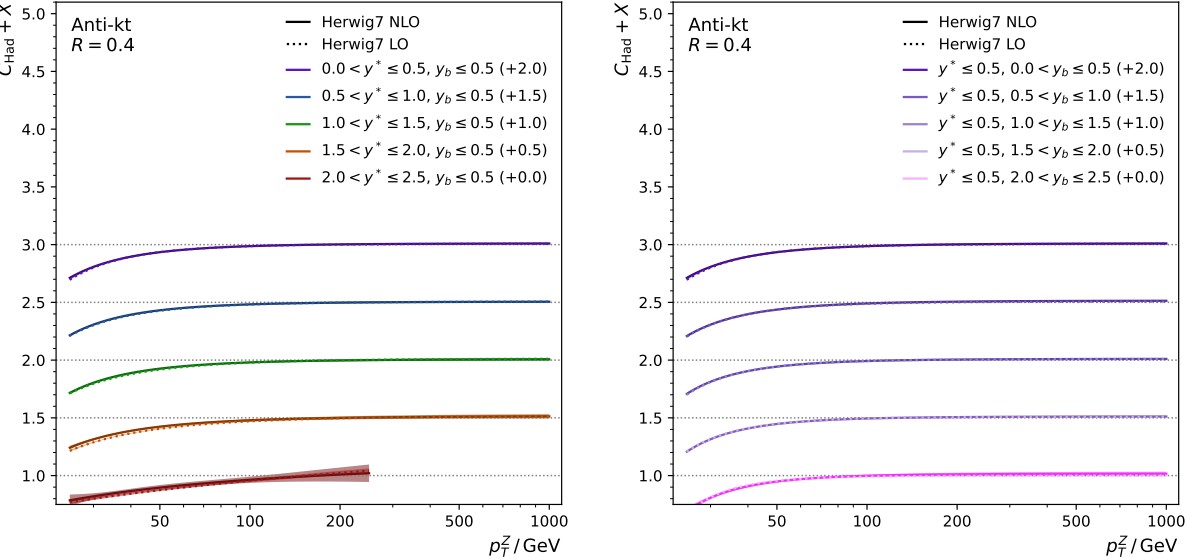

**Fig. 7**: Same as Fig. 4 but for hadronisation effects, $C_{\mathrm{Had}}$, only.

the upwards deviation from unity increases reflecting a higher share of MPI partons migrating into a jet. The magnitude of the increase towards small $p_{\mathrm{T,Z}}$ correlates with $y^*$ and decreases with increasing perturbative order. These observations indicate that the origin of the observed trends for the NP correction factors lies, at least partially, in the MPI model.

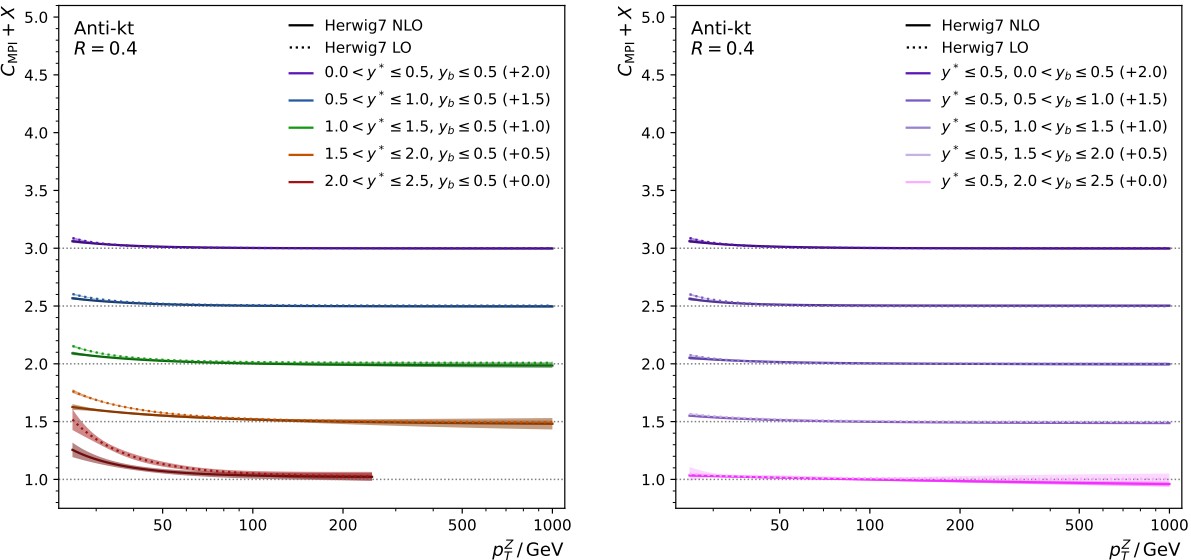

**Fig. 8**: Same as Fig. 4 but for MPI effects, $C_{\mathrm{MPI}}$, only.

# 5 Underlying Event Analysis

As observed in Sect. 4, both hadronisation and the UE each play a role for the NP corrections. Although these two effects cannot be completely decoupled, the trends with $y^*$ and the perturbative order in the NP correction factors noticed in Sect. 3 must mainly originate from MPI effects.

In this section, a closer look at the NP effects, focusing especially on MPI, is presented. For this purpose, observables of "Rick-Field-type" UE analyses that are typically used as sensitive observables in tuning and modelling studies [34–36] are utilised.

## 5.1 Analysis

We pre-select Z+jet events as in the previous sections in bins of $y_{\mathrm{b}}$ and $y^*$, however, with two modifications for the UE analysis: First, there is no minimum $p_{\mathrm{T,Z}}$ required, and secondly only charged particles are considered for the UE observables. Moreover, the $p_{\mathrm{T}}$ of each such particle must exceed 500 MeV in contrast to the default of 100 MeV set in RIVET. Then, the momentum of the lepton pair, i.e. the Z boson, is chosen to define the direction of the leading particle and henceforth the origin of the azimuthal angle $\phi$ for the UE analysis. All other charged particles, including

the ones from the jet, are categorised as belonging to one of three azimuthal regions, depending on the azimuthal angular difference $\Delta\phi$ relative to the leading particle, such that $-\pi < \Delta\phi \leq \pi$. Particles with $|\Delta\phi| \leq \pi/3$ are in the *towards* region, moving roughly forward relative to the leading particle. Particles moving in the direction of the recoiling jet, defined by $|\Delta\phi| > 2\pi/3$, are assigned to the *away* region. The remaining region with $\pi/3 < |\Delta\phi| \leq 2\pi/3$ is called the *transverse* region. Figure 9 illustrates the three regions relative to the (downwards-going) leading particle and the recoiling jet.

In a dijet, or minimum bias, event, the transverse region allows us to have a closer look at those particles that are furthest away from the leading, hard interaction. These particles are associated with the UE activity as it is observed, that beyond a certain threshold, the activity in the UE is more or less decorrelated from the hard interaction. In order to quantify the activity of the UE, two different observables are considered in bins of different "hardness" of the leading particle, characterised by the transverse momentum of the leading particle. Given a leading particle transverse momentum, the average number of charged particles $N_{\mathrm{ch}}$ or the sum of absolute transverse momenta $\sum p_T^{\mathrm{ch}}$ in either of the previously defined regions are considered as a measure for the activity in these regions.

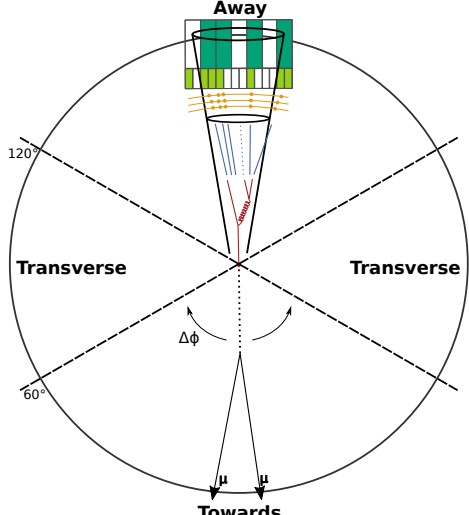

**Fig. 9**: Visualisation of the analysed phase space for the UE analysis. The away, towards, and transverse regions are defined relative to the direction in azimuthal angle $\phi$ of the dimuon system.

In dijet or minimum bias events, the activity in the towards and away region is also strongly associated with the transverse momentum of the leading particle. Hence, both observables tend to grow with increasing hardness of the event. The main driver of this is the parton radiation associated with the hardness of the event, which will mostly populate the towards and away region and can also spill over into the transverse region. If there is a portion of the event that is more or less uncorrelated to the hard event, as it is modelled in most MPI models, the activity in the transverse region should be dominated by the MPI activity or at least be the region that is least overshadowed from particles associated with the hard event.

In events with the Z boson as the leading particle, this is somewhat different. The hardness of the event is associated with the transverse momentum of the Z boson. With increasing transverse momentum of the Z boson the recoiling activity will mostly be in the away region while the region around the lepton pair, i.e. the towards region should be the region with the least activity from the UE.

In this work, it is now of interest whether the large NP corrections for relatively low transverse momentum of the Z boson, particularly for increasing values of $y^*$, are really related to the

activity from MPI. In these events, where the Z boson and the recoiling jet are getting closer and closer to the beam axis, there might also be a strong influence from parton radiation around the beam direction. Therefore, the UE observables are studied with different generators and for different matching and merging scenarios in order to discriminate activity due to hard radiation in the event from MPI activity.

## 5.2 Observations

To illustrate our findings, plots of the activity $\sum p_T^{\text{ch}}$ in the three different regions are shown. The results for $N_{\text{ch}}$ are qualitatively similar and therefore omitted in this discussion. These observables are derived from events generated with HERWIG7 and SHERPA with both LO and NLO accuracy of the hard Z+jet process. In addition, results obtained from the multileg-merged samples from SHERPA are compared.

The description of UE observables in inclusive Z+jet events with HERWIG7 has been considered before and is known to give a reasonable agreement with data [36–40]. We therefore focus on the UE observables in differential $(y_b, y^*)$ bins as defined in Sec. 2. In the following discussions, the outermost $y^*$ bin $(2.0 \leq y^* < 2.5)$ is omitted, because it is subject to large statistical fluctuations.

In Fig. 10 from top to bottom the results from LO ME event generation for the activity in the three regions away, towards and transverse are shown. In the left column $y_b$ is fixed to the central bin $(y_b < 0.5)$, while $y^*$ is varied. In the right column the respective results with $y^*$ fixed, while $y_b$ varies, are shown. It is apparent that the UE activity varies strongly with $y^*$ while there is no apparent variation with $y_b$. In the simplified back-to-back picture this means that the activity increases only when the central back-to-back hard event is tilted with $y^*$ but not when a central back-to-back hard event is boosted along the beam axis. The naïve expectation is that any contamination of activity from the beam or MPIs should occur in both situations likewise. Instead, the increased activity with $y^*$ in all regions hints at some relation between the hard event topology and the observed UE activity.

Considering the away region, it is found that the activity here is strongly correlated with

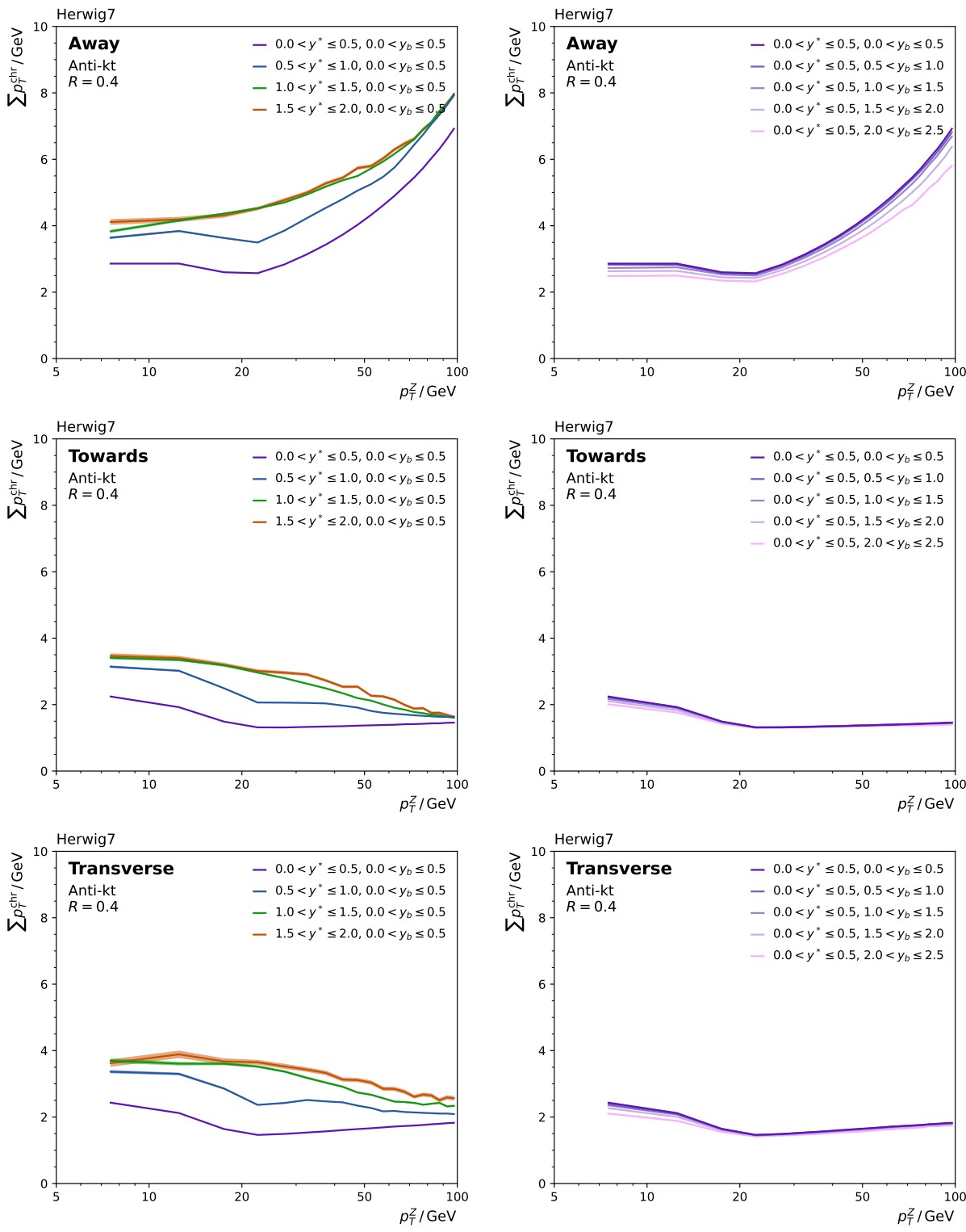

**Fig. 10**: Comparison of the sum of transverse momenta of all charged particles, $\sum p_T^{\mathrm{chr}}$, in bins of $p_{\mathrm{T,Z}}$, $y_{\mathrm{b}}$, and $y^*$ for the away (top), towards (middle), and transverse region (bottom row) as predicted by HERWIG7 using LO MEs for event generation up to particle level. The values are shown for a series of phase space intervals increasing in $y^*$ for $y_{\mathrm{b}} < 0.5$ (left) and increasing in $y_{\mathrm{b}}$ for $y^* < 0.5$ (right).

the transverse momentum of the Z boson and increases with $y^*$. This reflects mostly the activity in the recoiling jet which for increasing $y^*$ pulls in more particles from the beam direction. Also, we can clearly see that, as expected, the activity in this region increases significantly from around 20 GeV, which is the selection cut for the jet itself. Below 20 GeV in $p_{T,Z}$ the $p_T$ of the "balancing" jet must be compensated by other hadronic activity in the towards region. In fact, the hadronic activity in the towards and transverse regions decrease slightly from $p_{T,Z}$ = 7.5 GeV up to 20 GeV. Increasing $y_b$ at central $y^*$ in Fig. 10 (right column) does not exhibit significant changes in activity in any region.

Two cross-checks are performed. First, HER­WIG7 was used with a hard process at NLO instead of LO accuracy (not shown). Generally, the results are similar to the LO case. In particular, only a dependence on $y^*$ is observed but not on $y_b$. The activity in the transverse region is slightly lower at NLO than at LO. Secondly, SHERPA was employed with the ME at LO, NLO, and with multi-leg merging as shown for the transverse region in Fig. 11. As with HERWIG7 the UE activity in the transverse region strongly depends on $y^*$ but not on $y_b$, and decreases slightly with a more accurate simulation of the hard event.

Finally, in Fig. 12 the same NLO simulation with HERWIG7 with the full generation up to particle level is compared to two partial generation chains. More specifically, first, only the MPI is turned off (ME+PS+Had), and secondly both the MPI and hadronisation (ME+PS). In comparison to the full generation, mostly the overall normalisation changes suggesting the effect, that the MPI essentially adds activity to the event, but remains broadly uncorrelated with the hard event, as does hadronisation. In summary, hadronisation and MPI alone cannot fully explain the $y^*$ dependence and ME effects significantly contribute as well.

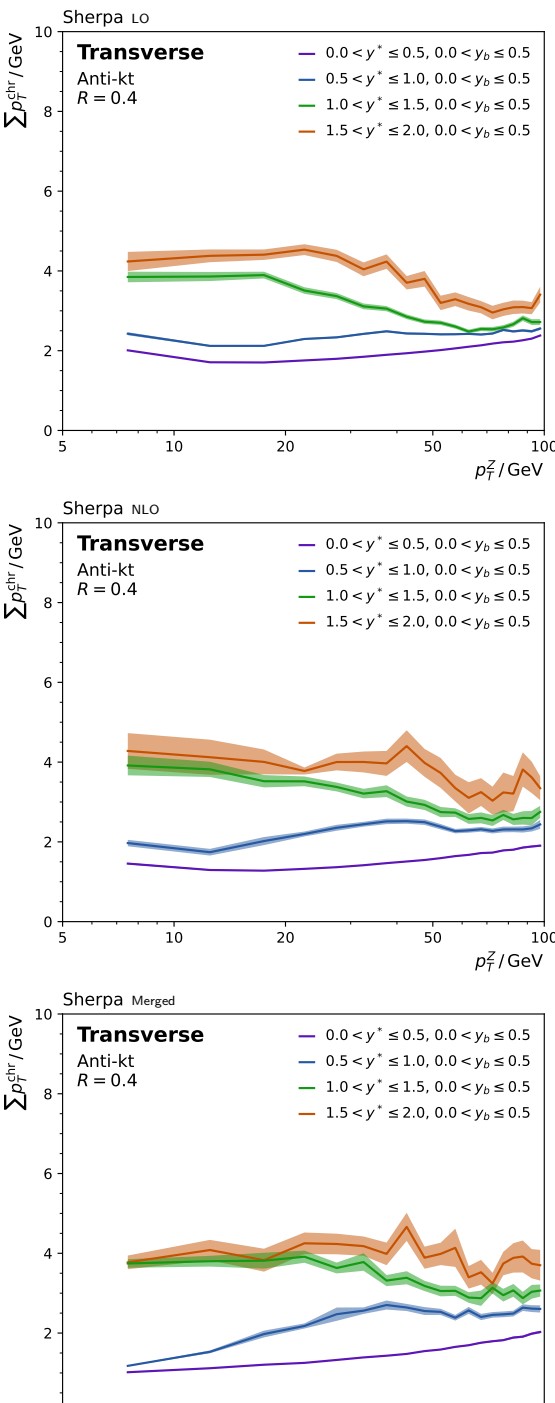

**Fig. 11**: Comparison of the sum of transverse momenta of all charged particles, $\sum p_T^{\mathrm{chr}}$, for central $y_b$ in bins of $p_{T,Z}$ and $y^*$ for the transverse region as predicted by SHERPA using LO (top), NLO (middle), and multileg-merged MEs (bottom) for event generation up to particle level.

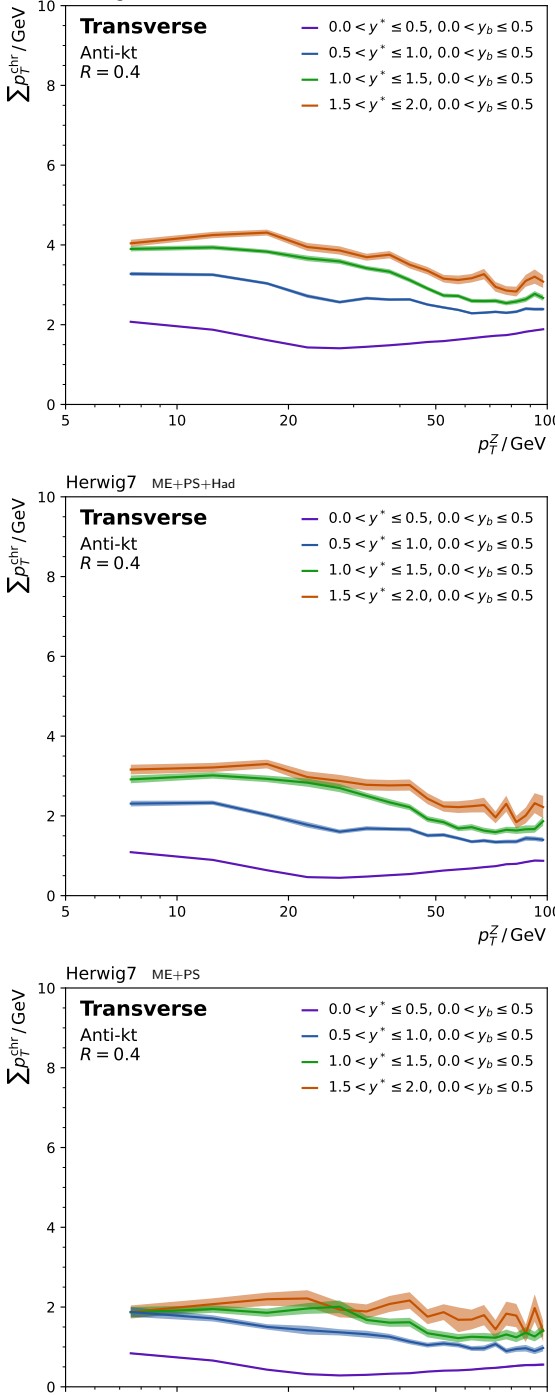

**Fig. 12**: Comparison of the sum of transverse momenta of all charged particles, $\sum p_T^{\mathrm{chr}}$, for central $y_{\mathrm{b}}$ in bins of $p_{\mathrm{T,Z}}$ and $y^*$ for the transverse region as predicted by Herwig7 using NLO MEs for event generation up to particle level with MPI (top), without MPI (middle), and up to parton level only, i.e. hadronisation and MPI off (bottom).

# 6 Summary and outlook

We find that nonperturbative (NP) corrections for Z+jet events in the triple-differential measurement of the cross section strongly depend on the scattering angle of the hard process in the centre-of-mass frame, $y^*$, but barely on the boost of the hard scatter, $y_{\mathrm{b}}$. Further inspection hinted at the Underlying Event (UE) simulation as being responsible for this extra activity. The generation with hadronisation effects switched off seemed to suggest this. However, looking into this effect with the help of a typical UE analysis in the same triple-differential binning has shown that a significant part of the extra activity must be of perturbative origin. The UE analysis shows the same strong dependence on $y^*$, even when the multiple parton interactions (MPI) as a model for the UE are switched off. Albeit this UE observable has not yet been measured triple-differentially, we find consistency between the simulation with the generators Herwig7 and Sherpa. In contrast, the NP correction factors for the dijet case do not exhibit this strong dependence on $y^*$.

In summary, we find that NP corrections, derived in the conventional way as factors, may not be entirely of NP origin. In addition, they are not universal among different hard processes. In order to clarify this situation, we recommend measuring the UE activity triple-differentially in Z+jet events.

# Acknowledgements

This work was supported by the German Federal Ministry of Education and Research (project funding numbers 05H2021, 05H21GUCC2, 05H21VKCCA), and by the Institute of Experimental Particle Physics and the Institute of Theoretical Physics at the Karlsruhe Institute of Technology, Germany. The authors acknowledge support by the state of Baden-Württemberg through bwHPC and the German Research Foundation (DFG) through grant no INST 39/963-1 FUGG (bwForCluster NEMO).

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

# Appendix A  Additional Material

Figures A1 and A2 show the NP correction factors as derived from HERWIG7 with LO and NLO MEs for dijet production versus $\langle p_T \rangle_{1,2}$ and $m_{1,2}$ for both jet size parameters, $R = 0.4$ and $0.8$, and all phase space intervals in $(y_b, y^*)$. The binning scheme for dijet mass is presented in Table A1, which, in contrast to $\langle p_T \rangle_{1,2}$, has not been extended to include results at lower $p_T$.

Lacking a physics-related motivation for the smoothing function of the NP corrections at low scales, it should be noted that the derived curves are valid only within the shown range and should not be extrapolated. In particular, when the inset of a steeper increase (or decrease) versus lower values of $\langle p_T \rangle_{1,2}$, $m_{1,2}$, or $p_{T,Z}$ is represented by only one bin at the low edge, statistical fluctuations may lead to exaggerated slopes.

Finally, Figures A3 and A4 show the NP correction factors as derived from HERWIG7 and SHERPA for all available perturbative orders of the MEs for Z+jet production versus $p_{T,Z}$ for both jet size parameters and all phase space intervals in $(y_b, y^*)$.

**Table A1**: The binning scheme in $m_{1,2}$ for the dijet analysis. Due to limited statistical precision, some bins are merged in the edge bins (E) as compared to the central bins (C) in the plane of $(y_b, y^*)$.

| Binning scheme | binning labels in $(y_b, y^*)$ plane |
|---|---|
| (cf. Fig. 2) | E |
| | E   E |
| | C   E   E |
| | C   C   E   E |
| | C   C   C   E   E |
| | bin edges in $m_{1,2}$ / GeV |
| Central (C) | 306, 372, 449, 539, 641, 756, 887, 1029, 1187, 1361, 1556, 1769, 2008, 2273, 2572, 2915, 3306, 3754, 4244, 4805, 5374, 6094 |
| Edge (E) | 372, 539, 756, 1029, 1361, 1769, 2273, 2915, 3754, 4805, 6094 |

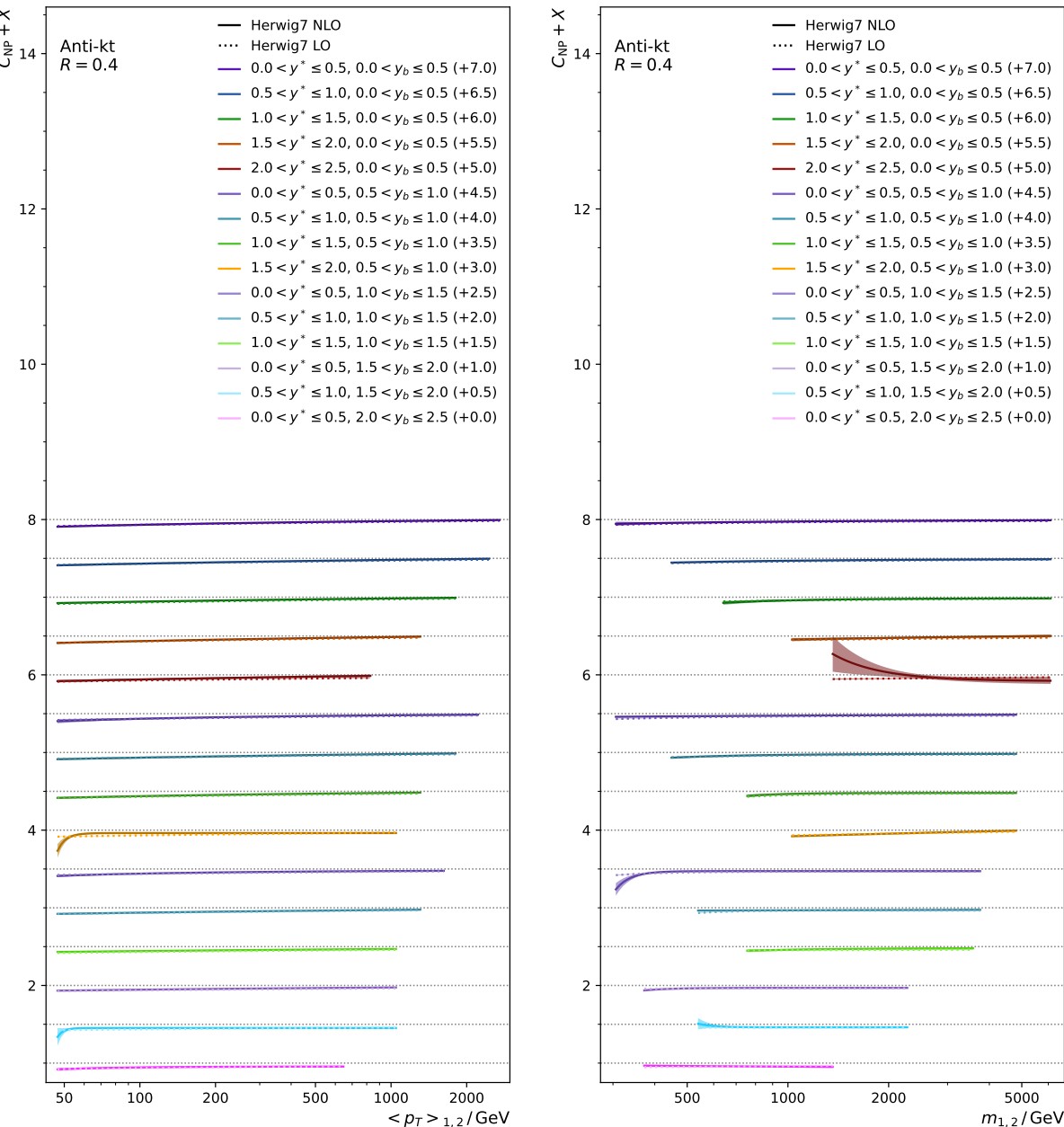

**Fig. A1**: $C_{\mathrm{NP}}$ vs. dijet $\langle p_{\mathrm{T}} \rangle_{1,2}$ (left) and mass $m_{1,2}$ (right) using LO (dotted) and NLO MEs (solid lines) in HERWIG7 predictions for anti-$k_{\mathrm{T}}$ jets with $R = 0.4$. The fitted curves are shown for all 15 defined phase space intervals in $(y_{\mathrm{b}}, y^{*})$ with bands indicating the statistical uncertainty. The factors are shifted with respect to unity by constant offsets.

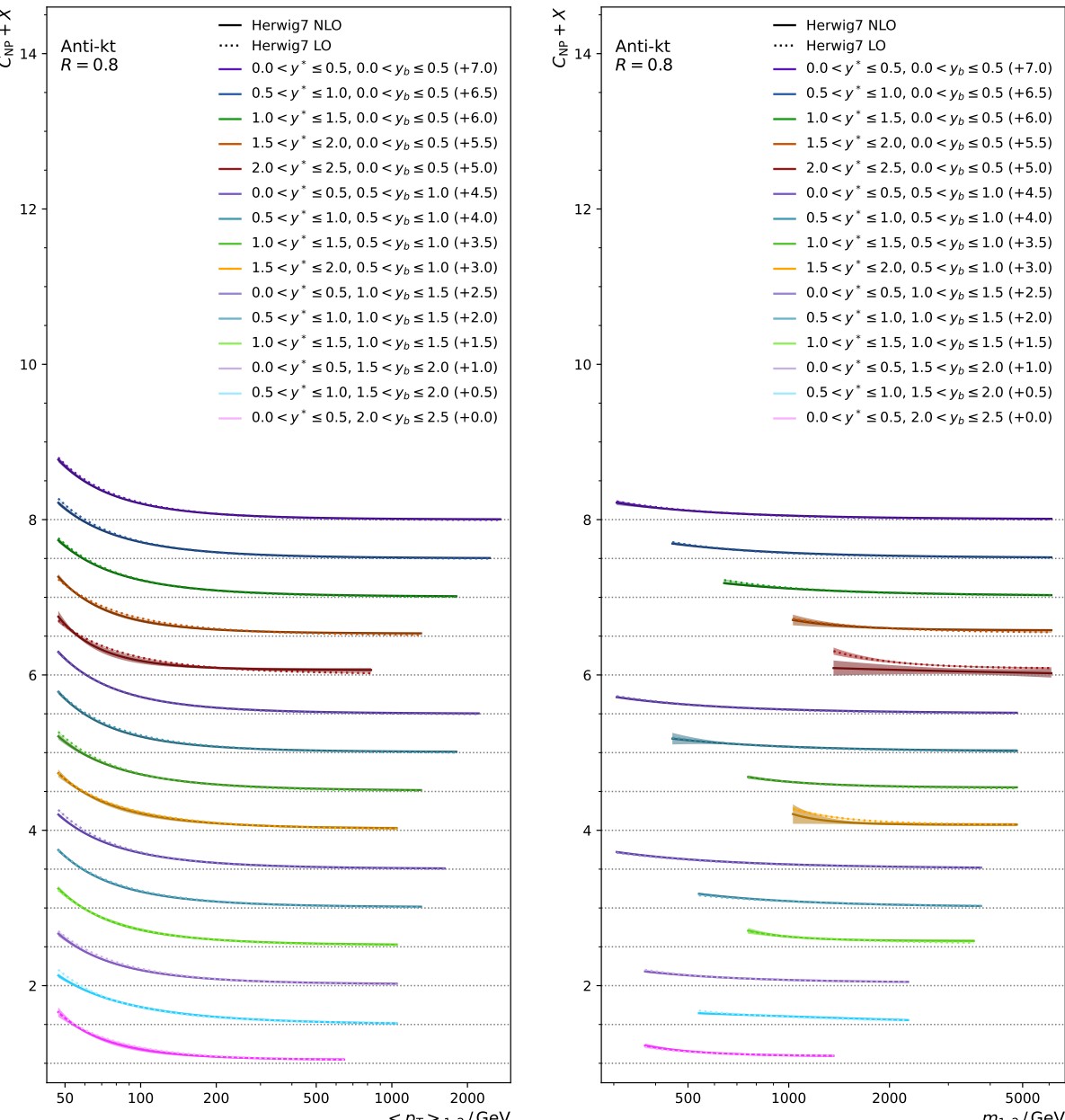

**Fig. A2**: Same as Fig. A1 but for anti-$k_{\mathrm{T}}$ jets with $R = 0.8$.

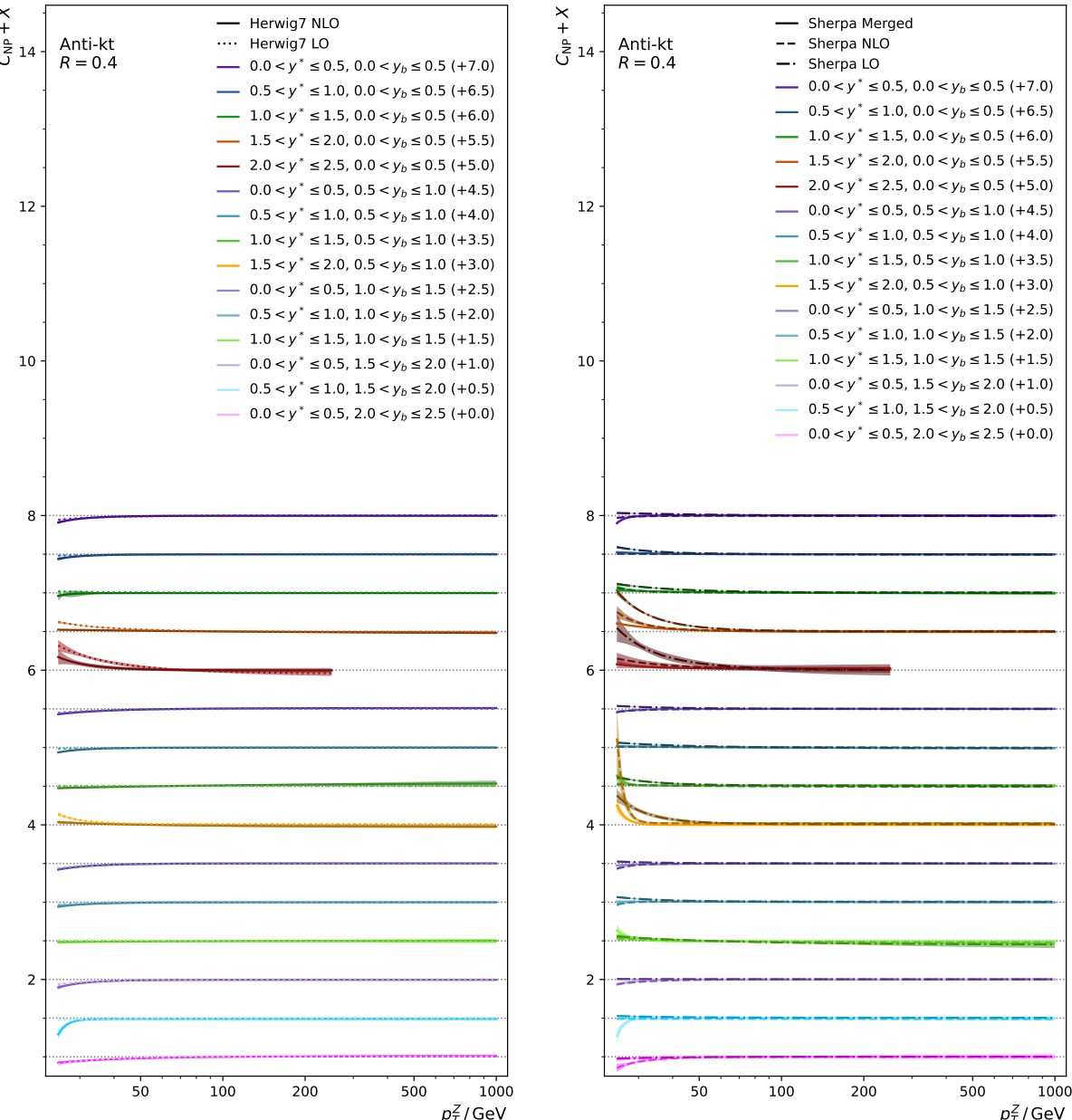

**Fig. A3**: $C_{\mathrm{NP}}$ vs. $p_{\mathrm{T,Z}}$ from HERWIG7 (left) using LO (dotted) and NLO MEs (solid lines), and from SHERPA (right) using LO (dash-dotted), NLO (dashed), and multileg-merged MEs (solid lines) for anti-$k_{\mathrm{T}}$ jets with $R = 0.4$. The fitted curves are shown for all 15 defined phase space intervals in $(y_b, y^*)$ with bands indicating the statistical uncertainty. The factors are shifted with respect to unity by constant offsets.

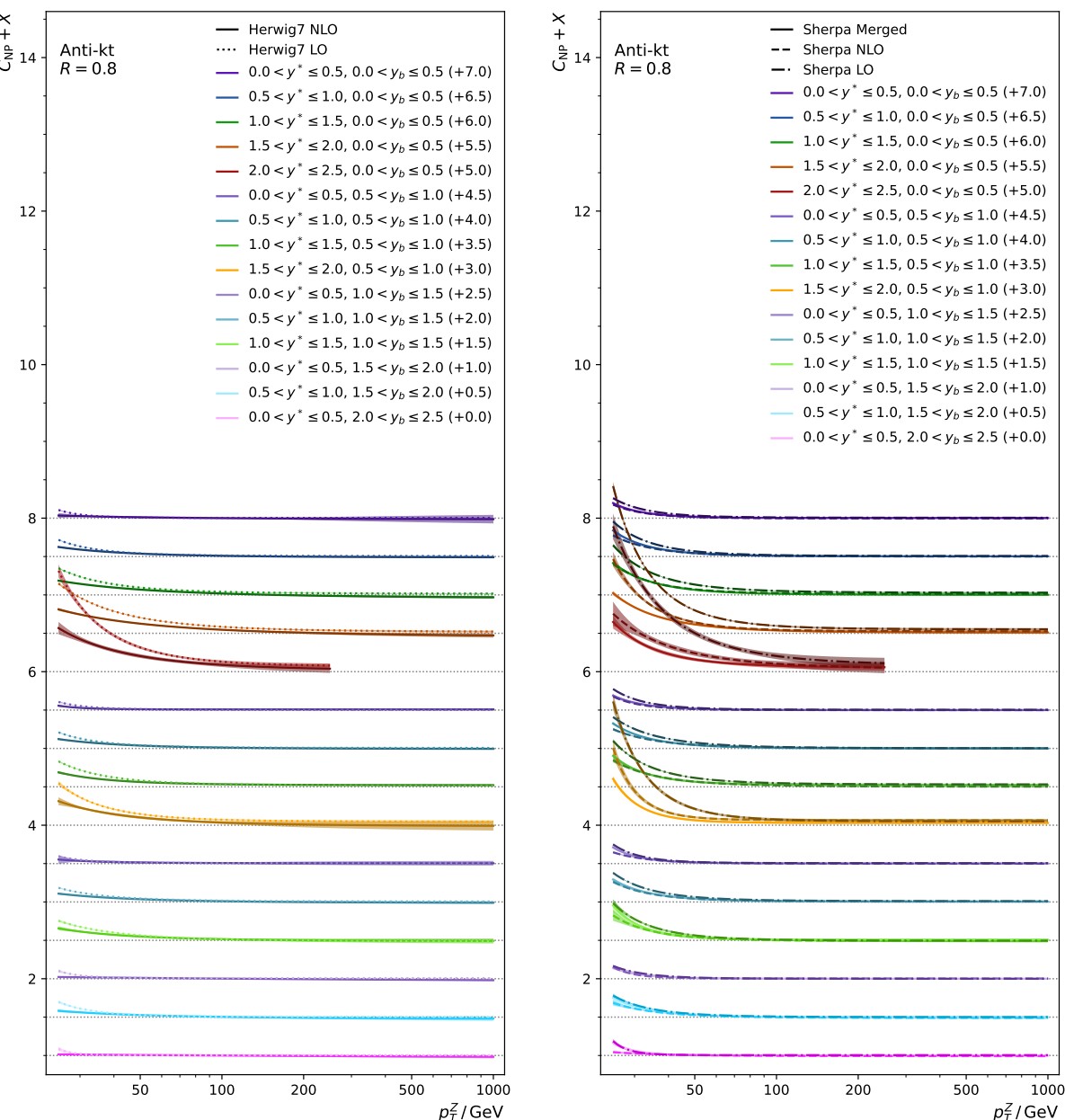

**Fig. A4**: Same as Fig. A3 but for anti-$k_{\mathrm{T}}$ jets with $R = 0.8$.