# Peer review of "Nonperturbative effects in triple-differential dijet and Z+jet production at the LHC"

_SciPost Physics_

## Round 2 · Referee Report · Anonymous (Referee 1) · 2025-12-20

Strengths

  1. Discusses non-perturbative (NP) corrections to important collider measurements in a clear and systematic way.
  2. Uncovers an interesting point about these NP corrections, that there are regions in which they are significant and appear to be non-universal.

Weaknesses

  1. Does not go as far in exploring possible explanations of the non-universal behaviour as one might hope (but this could be seen as a stimulation to the community to do so).
  2. Formulates the NP corrections as multiplicative rather than as a migration of jets up or down the pT distribution (which can be important since the two processes they compare have different pT slopes).
  3. Also formulates the NP corrections in a way that implies the parton shower of an event generator encapsulates the perurbative results and that the other components (hadronization and MPI) define the NP corrections.

Report

This is a good paper raising an interesting point, satisfies the journal criteria, and deserves to be published. I recommend some small changes and improvements first.

Requested changes

  1. There is no discussion of what the “Herwig LO”, “Herwig NLO”, “Sherpa LO” and “Sherpa NLO” results actually mean. Both programs have various options and running conditions and it is not at all clear just from these names which were used. Which of the parton shower algorithms, with which parameters, which MPI options and parameters, etc. Even the appellations “LO” and “NLO” are ambiguous, especially for the Z case - the LO process could be Z production or Z+jet production. And, in either case, what NLO matching algorithm/options are used? If the authors don’t want to clutter the main text with these details, an addition to the appendix would be ideal, but this really has to be spelled out in more detail for readers to make sense of the results.
  2. The paper takes a very traditional view of NP corrections as “everything that takes place in an event generator after the parton shower”. However, it is more commonly understood nowadays that the infrared cutoff of the parton shower is also the ultraviolet cutoff of the hadronization model and that one cannot separate them unambiguously - the cutoff dependence of the shower should be considered part of the NP correction. While I am not proposing that the authors change their approach at this stage, I do think it would be helpful to at least raise awareness of this issue, e.g. in the text after Eq. (6). At least to mention what the cutoff value is and how it is implemented (presumably as a pT cut) and remind readers that the parameters of the hadronization model are tuned in tandem with this value.
  3. It is common to think of NP corrections in terms of a migration (jets are shifted to higher or lower pt by radiation into or out of the cone), rather than a multiplicative rate correction. Again, I am not advocating a change of approach, but it would be helpful to mention this, and the reason for preferring a multiplicative correction, e.g. in the discussion around Eq. (7).
  4. In the discussion of section 5.1, there are a couple of small errors. It says that “In dijet or minimum bias events, the activity in the towards and away region is also strongly associated with the transverse momentum of the leading particle … The main driver of this is the parton radiation … “. Actually the main driver of the pT in the towards region is the momentum of the primary jet and in the away region the momentum of the other jet that recoils from it - the parton radiation mainly just serves to spread out this momentum, but the majority of it remains in the toward and away regions. And in the next paragraph, “The hardness of the event is associated with the transverse momentum of the Z boson” - this is somewhat true for the away region, which should contain the jet recoiling from the Z, but soft radiation into the transverse and toward region is determined by the scale of the hard process, which is something like the transverse mass - even Z events with zero pT produce soft radiation up to order mZ.
  5. Finally, a very very minor presentational thing - the page break in the middle of section 6 is very poor and, unless adding extra text to address the above points improves it anyway, I would suggest forcing the full-page figures to appear before starting the Summary and outlook section.

Recommendation

Publish (meets expectations and criteria for this Journal)

---

## Editorial Decision

in_refereeing